# Evolution Application of Two-Dimensional MoS_2_-Based Field-Effect Transistors

**DOI:** 10.3390/nano12183233

**Published:** 2022-09-18

**Authors:** Chunlan Wang, Yongle Song, Hao Huang

**Affiliations:** 1School of Science, Xi’an Polytechnic University, Xi’an 710048, China; 2Guangxi Key Laboratory of Processing for Nonferrous Metals and Featured Material, School of Resources, Environment and Materials, Guangxi University, Nanning 530004, China

**Keywords:** MoS_2_-FETs, logic and radio-frequency circuits, photodetector, biosensor, piezoelectric devices, synapses transistors

## Abstract

High-performance and low-power field-effect transistors (FETs) are the basis of integrated circuit fields, which undoubtedly require researchers to find better film channel layer materials and improve device structure technology. MoS_2_ has recently shown a special two-dimensional (2D) structure and superior photoelectric performance, and it has shown new potential for next-generation electronics. However, the natural atomic layer thickness and large specific surface area of MoS_2_ make the contact interface and dielectric interface have a great influence on the performance of MoS_2_ FET. Thus, we focus on its main performance improvement strategies, including optimizing the contact behavior, regulating the conductive channel, and rationalizing the dielectric layer. On this basis, we summarize the applications of 2D MoS_2_ FETs in key and emerging fields, specifically involving logic, RF circuits, optoelectronic devices, biosensors, piezoelectric devices, and synaptic transistors. As a whole, we discuss the state-of-the-art, key merits, and limitations of each of these 2D MoS_2_-based FET systems, and prospects in the future.

## 1. Introduction

The FET is voltage-adjustable electronic device that controls the output circuit current by controlling the electric field effect of the input circuit, which can work at very low current and voltage, and which can easily integrate on a piece of silicon and other substrates, so the FET has been widely used in large-scale integrated circuits [1]. The common FET device structure consists of a gate, source/drain, channel layer and dielectric. As is well known, the continuous miniaturization of silicon (Si)-based FETs has driven the exponential growth of integrated circuits for more than half a century. However, with the bulk thickness reduced to less than 10 nm, Si FETs exhibit a large number of attenuation phenomena of carrier mobility. This requires researchers to come up with a series of strategies to overcome this limitation, such as finding new materials, developing new structures, and improving processes [1,2].

Recently, 2D materials including graphene [3], black phosphorus (BP) [4], and transition metal dichalcogenides (TMDs) [5,6,7,8] have shown natural advantages for the further reduction in FET sizes due to their natural atomic-level thickness and surface without hanging bonds, for remedying the shortcomings of Si-based FETs. Table 1 summarizes the main electric performance of MoS_2_-based FETs and other mainstream 2D materials-based FETs. The graphene FET has ultra-high mobility. However, its I_on_/I_off_ ratio is lowest (typical below 10) owing to the lack of bandgap, and it is difficult to apply to logic electronics. The mobility of the BP FET is much higher than that of the MoS_2_ FET, but it is not stable, due to reaction with water and oxygen in the air to decompose. TMDs (MoS_2_, WS_2_, WSe_2_, etc.)-based FETs have an ultrahigh I_on_/I_off_ ratio and good mobility, which can be assembled into low-power electronics. Especially, MoS_2_ FETs have the advantages of higher mobility, higher I_on_/I_off_ ratio, and lower subthreshold swing, compared with other TMD FETs. Overall, MoS_2_ is the most suitable channel layer material for FETs, and MoS_2_ FETs are the promising candidate for downscaling electronics with a short channel, low thickness, small volume, fast speed, high sensitivity, light weight, etc.

Table 1 also shows that different MoS_2_ FETs have performed differently, because the natural atomic-level thickness and large specific surface area of MoS_2_ make the interface quality have a great influence on the MoS_2_ properties [2]. High-quality MoS_2_ preparation methods have constantly been investigated in recent years. The common preparation methods of MoS_2_ include Chemical Vapor Deposition (CVD), Metal–Organic Chemical Vapor Deposition (MOCVD), Atomic-layer deposition (ALD), and Vapor–Liquid–Solid (VLS) [18]. The properties of stripped MoS_2_ are better than those of CVD-prepared MoS_2_, because stripped MoS_2_ has high quality and less impurities, while the CVD process will bring a lot of impurities. In addition, a 6-inch uniform monolayer MoS_2_ can be grown by CVD in a short time [19]. MoS_2_ with the low surface energy can also be easily stripped and transferred, the size of which depends mainly on the size of the bulk MoS_2_, which can reach the centimeter level [20].

However, in MoS_2_ FETs, a high-quality MoS_2_ between the contact electrode and the dielectric layer will still be obtained at the contact interface and dielectric interface, which has further led to the different performance of MoS_2_ FETs [21]. Therefore, for improving the performance of MoS_2_ FETs, we mainly focus on optimizing the contact behavior [14], regulating the conductive channel [16], and rationalizing the dielectric layer [17]. This is also the subject of our review. Then, we discuss the applications of MoS_2_ FETs in key and emerging fields [21], involving logic and RF circuits [22,23], optoelectronic devices [24], biosensors [25], piezoelectric devices [26], and synaptic transistors [27] (Figure 1).

## 2. Performance Improvement Strategy of MoS_2_ FETs

### 2.1. Optimizing Contact Behavior

A metal electrode and traditional 2D electrode are usually directly deposited by chemical or physical deposition methods on MoS_2_ surfaces, which damages the contact interface, resulting in the Schottky barrier and Fermi-level pinning effect, between the MoS_2_/metal electrode, and the MoS_2_/traditional 2D semiconductor in MoS_2_ FETs [28], which could reduce electrical performance [29]. Therefore, it is very important to find suitable electrode materials to form ohmic contact and eliminate the Fermi pinning effect [30]. Recently, the van der Waals contact has been an indirect method of preparation into MoS_2_, which does not use direct chemical bonding, avoiding the damage of the deposition process and the diffusion at the contact interface, Here, we review the methods for obtaining ultra-low contact resistance in both 2D and non-2D contacts.

Shen et al. prepared a back-gated monolayer MoS_2_ FET, where bismuth, nickel, and titanium were used as electrodes to explore the contact barrier. The results showed that the Bi-MoS_2_ FET had the lowest contact barrier (123 Ω μm at a carrier density of 1.5 × 10^13^ cm^−2^), attributed to the suppression of metal-induced interstitial states by the semi-metallic bismuth contact with MoS_2_. This mechanism also contributes to good ohmic contact in other TMDs FETs, i.e., the I_on_/I_off_ ratios are as high as (10^7^) at low voltages (1.5 V) [21].

Samori et al. prepared an asymmetric Schottky diode by self-assembled monolayers (SAMs) of pre-functionalized gold electrodes, then transferring them to MoS_2_ by the dry method [14]. Figure 2a shows the MoS_2_ FETs structure, where the drain/source electrodes are 2, 3, 4, 5, 6-pentafluorobenzenethiol (PFBT)-functionalized electrodes and the source 4-(dimethylamine) benzenethiol (DABT) functionalized electrode, respectively. The output curve of the Schottky diode is shown in Figure 2b. The V_gs_ of the device is reduced from 90 to −30 V, and the rectification ratio reaches a maximum of 10^3^ at −30 V in Figure 2c. Figure 2d–f show the band structure of devices under different bias voltages. This strategy can be adjusted to control the electrode and different chemisorption SAMs functionalization to reduce (increase) the charge injection barrier, thus providing a strategy for the manufacture of asymmetric charge injection devices [14].

Recently, experiments have found that graphene materials as the electrodes of MoS_2_ FETs can also achieve good ohmic characteristics [31]. Yu et al. fabricated graphene/MoS_2_ heterojunction FETs, by adjusting the Fermi level of the graphene electrode to modulate the height of the Schottky barrier between heterojunctions, where the short-channel effect with a channel length less than 30 nm was successfully eliminated, and at the same time, the drain-induced barrier lowering of the device was 0.92 *v*/*v* and the I_on_/I_off_ ratio was as high as 10^8^ [31]. Kim et al. reported high properties of monolayer MoS_2_ FETs with a nitrogen-doped graphene (NGr) electrode [15], as shown in Figure 3. Compared to undoped graphene electrodes, the device current increased 214% and the field-effect mobility increased fourfold. Figure 3c,f illustrates that the Fermi level of MoS_2_ FET was improved by employing the NGr electrode. Therefore, this is one of the effective means to reduce contact resistance, which will provide a new idea for the development of high-performance devices.

Two-dimensional Mxenes contain transition metal carbides, carbonitrides, and nitrides, and have graphene-like surface structures and good electrical conductivity [32], which show great application potential in barriers, capacitors, and electrodes of electrochemical systems [33]. Dai et al. calculated the contact barrier of MoS_2_ FETs with Ta_2_C, Ta_2_CF_2_, and Ta_2_C(OH)_2_ as the electrode material, using density functional theory. The results showed that the N-type Schottky barrier was created by a Ta_2_C electrode. However, using a Ta_2_CF_2_ or Ta_2_C(OH)_2_ electrode can form ohmic contact, and the resistance of MoS_2_/Ta_2_C(OH)_2_ was 2 times smaller than that of MoS_2_/Ta_2_CF_2_. This study provides theoretical guidance for the application of MXene materials in MoS_2_ FETs [30]. Du et al. first studied the contact characteristics between MoS_2_ and M_3_C_2_(OH)_2_ (M = Ti, Zr, Hf) by first principles, which found that Ti and Hf are more suitable for ohmic contact as electrode materials. The subthreshold swing (SS) range of devices was 100~200 mV/decade, and the I_on_/I_off_ ratio was as high as 10^6^, in the sub-10 nm range [34].

To sum up, in the modification of the MoS_2_ FET electrode, the selection of an appropriate metal electrode or other excellent electrode materials can reduce the Schottky barrier, especially the combination of the above new two-dimensional material, and MoS_2_ can better achieve a high-performance, low-power-consumption FET, which will also be a significant attraction for the development of electronic devices in the future. In addition, 2D MoS_2_ could produce unique quantum effects by contacting with the metal, such as quantum transport, superconductors, and valley transport [35].

### 2.2. Boosting Conductive Channel

MoS_2_ with the good direct bandgap (~1.8 eV) when the bulk MoS_2_ is stripped to monolayer MoS_2_ sets the stage for MoS_2_ FETs with high mobility and high I_on_/I_off_ ratio [36]. As is well known, doping is one of the most prevalent techniques to regulate the band structure of semiconductor materials [37], which have both metal ion doping and nonmetal ion doping. An oxygen uniformly doped monolayer MoS_2_ can be prepared directly by in situ chemical vapor deposition on a 2-inch sapphire substrate. The results showed that the bandgap of MoS_2_ was regulated (from 2.25 eV (intrinsic) to 1.72 eV (heavily doped). The mobility of the MoS_2_-XOx FET was 78 cm^2^/Vs, and the I_on_/I_off_ ratio was 3.5 × 10^8^ [38]. Shi et al. developed a one-step CVD method to achieve the growth of centimeter-level monolayer MoS_2_ film [16]. Unidirectionally Fe-doped MoS_2_ domains (domain size up to 250 µm) were prepared on 2-inch commercial c-plane sapphire, which achieved very low contact resistance (≈678 Ω µm) and good ohmic contact with the electrode, as shown in Figure 4a–c. MoS_2_ FETs obtained electron mobility (54 cm^2^V^−1^s^−1^ at room temperature, 94 cm^2^/Vs at 100 K, and I_on_/I_off_ ratio (10^8^)). The electron mobility of monolayer Fe-MoS_2_ was decreased with increasing temperature, as observed in Figure 4d,e, which indicated that it can inhibit the scattering of impurities. Figure 4f shows the energy barriers of pristine MoS_2_ and Fe-MoS_2_.

Wang et al. first synthesized a Ta-doped p-type monolayer MoS_2_ by NaCl-assisted CVD, which has the advantages of large area, controllability, high quality, and controllable doping concentration [39]. With the addition of Ta, MoS_2_ FET devices showed bipolar properties and changed from N type to P type with increasing concentration. The p-type MoS_2_ active layer was applied with heavy niobium doping by mechanical stripping on the Si substrate. Then, the p-type MoS_2_ FETs with the Pt electrode obtained a 0.13 eV contact barrier, output current of −10 µA, and drain voltage of −1 V when the channel length was ~1 µm [40]. Han et al. reported a strategy for the controllable transformation of n-type MoS_2_ into p-type MoS_2_ by low energy (100 eV) He^+^ irradiation, as shown in Figure 5. Through theoretical calculation and characterization, it was found that this method increases the band size through the migration of the topmost S atom, and electron capture transforms n-type MoS_2_ into p-type MoS_2_ [41].

Doping can effectively improve the performance of MoS_2_ films, but in the case of thin atom thickness or few layers, traditional doping strategies such as ion implantation or high-temperature diffusion could not improve the performance, because these methods tend to destroy the lattice and produce defects. Therefore, it is particularly important to find new doping methods, which provides inspiration for future research and development of more processes suitable for less layer MoS_2_ doping.

### 2.3. Rationalizing Dielectric Layer

As for the MoS_2_ FET, the surface accumulation of charges and charge traps on the dielectric are big issues leading to high leakage current, which is observed by using a scanning tunneling microscope [42,43]. Therefore, an appropriate dielectric layer with high permittivity should be used to effectively reduce the influence of impurity scattering at the interface on carrier transport in the channel layer, while avoiding the possible short-channel effect and reducing gate leakage current [44,45]. It is noted that a high-K dielectric layer (HfO_2_, ZrO_2_, HfZrO, etc.) can effectively shield the scattering of charged impurities and boost the gate control ability on the channel carriers, thus effectively improving the electrical performance of MoS_2_ FETs [46,47,48,49]. However, the high-K dielectric layer will also generate trap charges and surface optical phonons to offset its advantages [49].

To solve these problems, Song et al. found that adding Al to the ZrO_2_ dielectric layer in a certain proportion (Zr:Al = 1:1) could effectively improve the electrical performance of MoS_2_ FETs, due to the moderate Al reducing the oxygen vacancy, and optimize leakage current, as well as improve the interface state density [50]. Zhao et al. demonstrated HfO_2_ with treatment by NH_3_ plasma and added Al_2_O_3_ as a high-K dielectric layer of the top-gate MoS_2_ FETs. That significantly reduced the leakage current and provided a high carrier mobility ~87 cm^2^/Vs, I_on_/I_off_ ~2.1 × 10^7^, and SS of 72 mV/dec [51]. Li et al. reported MoS_2_ FETs with the self-limiting epitaxy technology, employing a monolayer molecular crystal of perylene-tetracarboxylic dianhydride as a buffer layer, graphene as a gate, and HfO_2_ as a gate dielectric layer, which obtained good dielectric properties, low leakage current, and high breakdown power. The device has SS ~73 mV/dec, I_on_/I_off_ > 10^7^, and is not affected by short-channel effects [52]. Liao et al. designed high-performance MoS_2_ FETs with the double-layer gate dielectric structure, as shown in Figure 6a,b. A high-K vinylidene fluoride-trifluoroethylene (P(VDF-TrFE)) dielectric is used to provide high carrier concentrations, and a low-K polymethyl methacrylate (PMMA) dielectric provides device stability, as shown in Figure 6d,e. MoS_2_ FETs with low operating voltage, no hysteresis, and high stability were achieved through combining high-K and low-K dielectrics, as shown in Figure 6f [17].

Therefore, an excellent dielectric layer is important to obtain nonhysteresis, low voltage, and stable operation for MoS_2_ FETs. A high-K dielectric can effectively suppress the scattering of charge impurities and the other problems. However, when the 2D MoS_2_ material as the channel layer is very thin, the interface state problem will be more significant. Therefore, finding suitable single-layer/multi-layer dielectric materials and their high-quality deposition methods, or adding possible buffer layers, etc., can play a role in reducing these negative effects of monolayer MoS_2_ FETs.

## 3. Logic and Radio-Frequency Circuits

### 3.1. Logic Circuits

Logical computation is an important part of a computer. Silicon complementary metal oxide semiconductor (CMOS) circuits are widely used at present, and they have complementary features to ensure data transmission will not produce problems, such as loss threshold voltage, accurate logic transmission, high I_on_/I_off_, and easy integration. The application of graphene in logic devices is limited by the zero bandgap. Therefore, the MoS_2_ with a suitable bandgap provides a basis for the future application in logic computing devices [53]. Simulations revealed that monolayer MoS_2_ FETs showed a 52% smaller drain-induced barrier lowering, and a 13% smaller SS, than the 3 nm thick-body Si FETs at a channel length of 10 nm [54]. Therefore, MoS_2_ FETs are expected to improve the performance, and obtain new functions of devices in the field of electronics and display technology [55].

For voltage switching and high-frequency operation, logic devices need to achieve some performance, such as a high I_on_/I_off_ ratio (>10^3^) and moderate mobility [18]. Ang et al. prepared a top-gate MoS_2_ FETs. The monolayer MoS_2_ was grown on sapphire substrates by CVD and employed HfO_2_ as a high-K dielectric layer. The inverter was fabricated by direct-coupled FET logic technology to obtain a high voltage gain ~16. When V_DD_ = 3 V, the total noise margin was 0.72 V_DD_ [56]. Pan et al. proposed a double-gate MoS_2_ FETs to overcome the difficulty in regulating the threshold voltage and SS under the condition of a single gate. The device obtained an ultralow SS value of 65.5 mV/dec in the large current range above 10^4^, when the dual gate was operating simultaneously in the inverter [57].

Multiple inverters require FETs with a stable resistance ratio to ensure constant output voltage. Kim et al. prepared cross-type FETs and the WSe_2_/MoS_2_ PN heterojunction surface was treated with PMMA-co-PMAA, so the channel current and anti-bipolar transistor region characteristics improved. Figure 7c,d shows that PMMA-co-PMAA can increase the current and ensure the stability of the inverter by doping effect-inducing charge transfer. Applying a cross-type p−n heterojunction WSe_2_/MoS_2_-based FET into a ternary inverter, three stable logic states of 1, 1/2, and 0 were realized [22], as shown in Figure 7a,b.

Zhou et al. fabricated MoS_2_ FETs with a double-gate structure and double surface channel, where the top gate and the back gate serve as two input signals. Logic (OR, AND) was successfully implemented in a single cell [58]. MoS_2_-based FETs will replace the traditional silicon device to achieve a higher degree of integration of logic devices, but only simple logic can be achieved at present. On the one hand, the development of P-type MoS_2_ FETs still has a low performance and complex process problems, so its application in CMOS circuits still has a long way to go. On the other hand, mass production of large-size monolayer MoS_2_ is also a difficulty. However, we still believe that MoS_2_-based FETs will be applied to complex and scalable large-scale integrated circuits in the future.

### 3.2. Radio-Frequency Circuits

With the advent of the 5G communication era, the position of RF devices is increasingly prominent and comparable to the importance of logic devices and memory devices. Facing the problem of bandwidth growth, promoting the working frequency of RF transistors is inevitable [53]. The maximum frequency of oscillation (f_max_) based on III-V materials has exceeded 1 THz [59]. To further improve the operating frequency and bandwidth, optimizing the device processes or finding materials with higher mobility such as graphene is an effective strategy. Although the RF devices made of graphene can reach the operating frequency of mainstream RF devices, the gain cannot be further improved, due to the restriction of the zero bandgap. However, MoS_2_ with a sizable bandgap can solve this bottleneck effectively [60].

Cheng et al. demonstrated a high-performance MoS_2_ RF device on a flexible substrate with an intrinsic cut-off frequency of 42 GHz and f_max_ up to 50 GHz, and an intrinsic gain over 30 [61]. Wu et al. reported a microwave circuit with a MoS_2_ self-switching diode. Ten layers of MoS_2_ were grown on an Al_2_O_3_/Si substrate to realize the audio spectrum of an amplitude-modulated microwave signal in the 0.9~10 GHz band [62]. Modreanu et al. fabricated RF transistors with a double-layer MoS_2_ channel structure, and the RF performance of the device was improved by electrostatic doping of the back gate. For MoS_2_ RF transistors with a 190 nm gate length, at V_BG_ = 3 V, the external natural cutoff frequency was 6 GHz, the internal natural cutoff frequency was up to 19 GHz, and the maximum oscillation frequency was 29.7 GHz [63].

Among the flexible electronic devices, the f_T_ and f_max_ of MoS_2_ RF transistors were higher than or comparable to those based on flexible electronic materials such as silicon film and InGaZnO [59]. Gao et al. transferred the bilayer MoS_2_ onto a flexible polyimide substrate to prepare high-performance RF transistors. The different gate lengths (0.3 µm, 0.6 µm, and 1 µm) affected the performance of MoS_2_ RF transistors. It was found that f_T_ and f_max_ increased with the decrease in the gate length, and with the gate length of 0.3 µm, the external f_T_ was 4 GHz and f_Max_ was 10 GHz [64].

Dresselhaus et al. formed a (1T/1T’-2H) phase heterostructure of MoS_2_, and a Schottky diode flexible rectifier was fabricated by ohmic contact with palladium and Au electrodes, as shown in Figure 8d. Figure 8a–c show that the rectifier voltage increased with the increase in RF power, the maximum power efficiency could reach 40.1% at the 2.4 GHz band, and f_T_ was 10 MHz. A radio-frequency energy collector was prepared by integrating with a flexible Wi-Fi antenna, as shown in Figure 8e. The radio-frequency energy collector at 2.5 cm produced an output voltage of 250 mV at 5.9 GHz. These provided a universal energy-harvesting building block, which could also be integrated with various flexible electronic systems [65].

Most MoS_2_-based RF transistors have been concentrated in bilayer structures because bilayer MoS_2_ typically has higher mobility and saturation speed than single-layer MoS_2_, providing a better power gain and cutoff frequency for the device. However, the high-frequency performance of MoS_2_ RF transistors is still lower than that of silicon transistors. Therefore, it is important to optimize the device structure or process (such as optimizing gate structure and edge contact). In addition, MoS_2_ with excellent mechanical properties can be used to manufacture flexible devices and, combined with RF, logic, and other fields, can achieve flexible electronic systems such as wearable devices, flexible sensors, and medical equipment.

## 4. Photodetectors

A photodetector is a device that converts photons into current by the photogenerated voltage effect of semiconductor materials. When a light source with strong radiation energy illuminates a semiconductor, the semiconductor material will absorb photons and generate electron–hole pairs to generate photocurrent [66]. MoS_2_ has a wide bandgap, layered structure, strong photoluminescence characteristics, and excellent mechanical properties. Photodetectors made of MoS_2_ have strong light response characteristics in the terahertz, mid-infrared, visible light, and near-infrared ranges [67]. Widely used photodetector performance indicators include responsivity, time response, specific detection rate, and spectral selectivity [68].

Zhang et al. developed a method for rapid synthesis of multilayer MoS_2_ films from 1 L to more than 20 L with good quality using NaCl as a promoter. Then, a monolayer-to-multilayer MoS_2_ photodetector was constructed, as shown in Figure 9a. Figure 9b,c illustrates the output curve of the monolayer–multilayer MoS_2_ heterojunction device and the outstanding rectifying ratio (10^3^). Figure 9d depicts the band diagram of the monolayer MoS_2_ under high performance. In order to explore photoresponsivity, the transfer characteristic curve (Figure 9e) of the device under light and dark conditions and the light response ability under different voltages were investigated. Figure 9f shows that the device could achieve a maximum sensitivity of 10^4^ A/W at 0 V (V_GS_) [69].

Wu et al. prepared a photodetector by combining MXene nanoparticles with a 2D MoS_2_ in a hybrid plasma structure. This strategy could improve the optical response of MoS_2_ and make it more sensitive to visible light. The experimental results showed that the response rate of the device was 20.67 A/W, the detectivity was 5.39 × 10^12^ Jones, and the external quantum efficiency was up to 5167% [70]. Wang et al. grew a gate dielectric heap (Al_2_O_3_/HZO/TiN) on a Si substrate and transferred multilayer MoS_2_ to prepare an ultra-sensitive negative-capacitance MoS_2_ phototransistor. The HZO film could significantly enhance the optical gating effect, suppress dark current, and improve the ratio of light to dark current through a ferroelectric local electrostatic field and the ferroelectric NC effect. The experiment demonstrated that the prototype device had a high detection rate of 4.7 × 10^14^ cm Hz^1/2^W^−1^ and a high response rate of 96.8 AW^−1^ at low operating voltages of V_ds_ = 0.5 V and V_g_ = 1.6 V at room temperature [71]. Walila et al. grew a 3.5 µm GaN on a sapphire substrate, and the MoS_2_ was mechanically stripped as the active layer. Figure 10a,b show the highly sensitive photodetector with gold and chromium as the electrode. Compared with bare GaN photodetectors, the responsivity and EQE of the photodetectors improved by 5 times, as shown in Figure 10d,e. Under the condition of 1 V bias and 365 nm excitation at 1 mW/cm^2^, the highest response rate and EQE were obtained, which were 1.8 × 10^4^ A/W and 6.19 × 10^6^%, respectively [24]. The GaN/MoS_2_ heterojunction laid a foundation for wide bandgaps and excellent photoelectric performance.

At present, the MoS_2_ photodetector mainly focuses on two aspects: the change in layer number and the construction of a heterojunction. The monolayer MoS_2_ has a low absorption surface and low quantum yield, which limits its development in photoelectric devices. In contrast, the multilayer MoS_2_ has a small bandgap and can improve the absorption efficiency. Especially, the MoS_2_-based heterojunction photodetector could achieve faster charge transfer and optical response. Optoelectronic devices require trade-offs between responsiveness and response time to meet practical needs. Although MoS_2_-based photodetectors have some difficulties, they still have great advantages in flexible photodetectors and integrated nano-optoelectronic systems.

## 5. Biosensors

Biosensors are widely used in clinical and disease treatment as a powerful tool to detect biochemical processes. The great demand for detection has promoted the development of new nanomaterials as a sensing platform [72,73]. The global spread of COVID-19 has warned people to protect their own health, and biosensors with rapid, real-time, and accurate detection can effectively contain the spread of the virus and remind people to treat themselves in a timely manner [73,74]. MoS_2_ attracted new interest with the multidimensional structures and structure-dependent unique electronic, electrocatalytic, and optical properties [75]. Therefore, MoS_2_ has been widely applied in biosensors that can detect DNA, proteins, metal ions, and other compounds [76].

Arshad et al. modified Au nanoparticles on MoS_2_ nanosheets to prepare bottom-gate FETs for the detection of the low-concentration C-reactive protein. The detection limit and sensitivity of BG-FETs were 8.38 fg/mL and 176 nA/g·mL^−1^, respectively [77]. Dai et al. prepared MoS_2_ on a 300 nm SiO_2_/Si substrate and functionalized MoS_2_ by combining five different DNA sequences into a DNA tetrahedron, as shown in Figure 11a. Figure 11b shows that the biosensor is extremely sensitive to the target protein (prostate-specific antigen, PSA). In phosphate-buffered brine, the detection limit was 1 fg/mL and the linear range was 1 fg/mL~100 ng/mL. Figure 11c shows that the I_ds_ decreases as the concentration increases, and bovine serum albumin (BSA) is used as an interference signal for comparison of detection effects. Figure 11d shows that the biosensor is extremely sensitive to the target protein (prostate-specific antigen, PSA) [25].

He et al. prepared a surface plasmon resonance (SPR) biosensor with Au nanoparticles-modified MoS_2_ nanoflowers for IgG detection. The flower-like structure can provide more active sites for metal particles to react with target substances. The sensitivity of the MoS_2_-Aunps-modified sensor was 0.0472 nm/(μg/mL), which is about 3 times higher than that of an unmodified sensor (0.016 nm/(μg/mL)). The limit of detection of IgG was reduced 2.7 times (from 0.16 to 0.06 μg/mL) [78]. Kim et al. prepared MoS_2_ biological FETs with a nanoporous structure. Nanoporous structures could increase the edge area, using block copolymer photolithography. The surface area of the nanocycle was selectively functionalized by the newly formed suspended groups at the edge of the nanocycle. The biosensor exhibited superior detection performance in human serum and artificial saliva and resulted in a limit of detection of 1 ag/mL for cortisol [79].

This section mainly introduces the MoS_2_-based biosensor and the biocompatibility of the MoS_2_ material with biological cells, and the portability, sensitivity, and low power consumption of FETs have been widely studied and applied in biological monitoring. During the background of COVID-19, the development of wearable biosensors for rapid monitoring has become an urgent need.

## 6. Piezoelectric Devices

Piezoelectric properties exist in materials with centrosymmetric fractures. When strain was applied, the center of gravity of the cation and anion did not coincide, which resulted in a voltage potential at the interface between the semiconductor and the metal [80]. The mechanical flexibility and piezoelectric and photoelectric effects of MoS_2_ materials can meet the needs of pressure sensors, micro-electro-mechanical systems, and active flexible electronic devices [81]. A layered MoS_2_ material could be modified to obtain piezoelectric properties and could be applied to piezoelectric nanogenerators [82]. The piezoelectric effect can be used to collect micro-mechanical energy and convert it into electrical energy [83].

Hone et al. first reported the piezoelectric properties of monolayer MoS_2_ in 2014 [84]. Kim et al. prepared a monolayer MoS_2_ piezoelectric nanogenerator (PNG) by sulfur vacancy passivation. The output peak current and voltage of the PNG monolayer MoS_2_ nanoflakes treated by S increased by 3 times (100 pA) and 2 times (22 mV), respectively. In addition, the maximum power increased by nearly 10 times [85]. Hu et al. fabricated a single-layer butterfly MoS_2_ piezoelectric device on a polyethylene terephthalate substrate. Figure 12a,d shows the piezoelectric properties of the MoS_2_ single crystal (SC-MoS_2_) and MoS_2_ with grain boundaries (GB-MoS_2_) under external strain. Figure 12b,c shows that under the action of external stress, the current value generated by the direction of “armchair” was higher than that of the direction of “Zigzag”. Figure 12e,f shows that the GB-MoS_2_ current density is higher than that of SC-MoS_2_. It was found that the piezoelectric effect induced by the grain boundary (~ 50%) could be applied to a self-powered sensor to monitor changes in human blood pressure [26].

Xue et al. reported a new self-powered NH_3_ sensor, which employed monolayer MoS_2_ materials on PET and covered polydimethylsiloxane films, and deposited an Au electrode. The sensor could be worn on different parts of the body and was responsive with a fast response/recovery time of 18 s/16 s [86]. Willatzen et al. first calculated the piezoelectric coefficient of 3R-MoS_2_. Here, the 5-layer 3R-MoS_2_ structure had the highest piezoelectric constant in all MoS_2_ multilayer structures. The maximum piezoelectric constant was about 13% higher than that of the monolayer MoS_2_ structure [87].

The discovery of piezoelectric effects could utilize a lot of neglected energy. At present, the application of MoS_2_ piezoelectric properties is not mature enough. It is necessary to systematically study the piezoelectric properties, piezoelectric coefficients, and deformation direction of MoS_2_ with different layers to help develop high-performance piezoelectric devices. Hence, MoS_2_ piezoelectric sensors will realize their great potential in nanoscale electromechanical systems, micro-flexible wearable devices, and other fields in the future.

## 7. Synaptic Transistors

After the concept of artificial intelligence appeared, neuromorphic electronics, which simulate human brain function and information processing, have been proposed as an effective method to solve complex data processing problems [88]. The foundation of this technology is to make artificial synapses, which have low power consumption, small size, and simple structure. A MoS_2_ material is appropriate for the construction of artificial synapses due to its excellent electrical properties and optical response. At present, a MoS_2_-based artificial synapse has been applied in memory devices, programmable logic circuits, and other fields.

Roy et al. fabricated a MoS_2_ vertical synaptic transistor with a titanium and Au electrode. The device exhibited extremely low cycle-to-cycle variability and device-to-device variability and stability in the SET voltage and RESET power distributions. The results showed that there are 26 different conductance states in the device, and each state is maintained for at least 300 s. These devices maintained a consistent on/off ratio during the 1000 DC SET–RESET cycles [89]. Wang et al. proposed a phototransistor based on a MoS_2_/graphene heterostructure and an integrated triboelectric nanogenerator to simulate mechanical photon artificial synapses. Synaptic plasticity can be realized by modulating the channel conductivity of the phototransistor by regulating the mechanical displacement of a TENG. The simulation results showed that the image recognition accuracy of the artificial neural network was improved by 92% with the help of mechanical plasticizing [90].

Im et al. fabricated multilevel memory based on van der Waals heterostack (HS) N-MoSe_2_/N-MoS_2_ FETs and extended it to synaptic memory. Figure 13a shows that the synaptic stacked channel FET was used to simulate a biological synapse. Figure 13b,c illustrates that using a voltage pulse could simulate synaptic behavior. HS memory utilized the capture/de-capture phenomenon induced by V_GS_ for programming/erasing functions. Due to the existence of a heterojunction energy barrier between MoS_2_ and MoS_2_, it could maintain a long retention time of 10^4^ s. Based on the P-D characteristics of the device under multiple 60 s short V_GS_ pulses, the simulated recognition rate could reach 94% on average [27], as shown in Figure 13.

The successful fabrication of artificial synapses based on MoS_2_ has proved the possibility of its application in non-von Neumann computing. MoS_2_ synaptic transistors with high performance, low power, and large-scale integration characteristics will be widely used in the brain-like chip, logic circuit, and simulated artificial neuromorphic system. However, there are still significant challenges in the fabrication technology and the structure of MoS_2_ synaptic transistors.

## 8. Conclusions

Two-dimensional MoS_2_ FETs have attracted wide and in-depth attention as a suitable candidate for optoelectronic devices and next-generation large-size flexible electronics. This benefited from the natural atomic layer thickness and large specific surface area of MoS_2_, but it can impact the influence quality, which mainly contains a contact interface and dielectric interface, and they both can then influence the performance of the MoS_2_ FET. Thus, in order to obtain higher-performance MoS_2_ FETs, it is necessary to focus on its main performance improvement strategies, including optimizing the contact behavior, regulating the conductive channel, and rationalizing the dielectric layer.

To optimize the contact behavior of MoS_2_ FETs, this paper reviewed metal, graphene, Mxene, and other new electrodes materials contacting with MoS_2_, which can adjust the Fermi level and reduce the Schottky barrier. Self-assembling monolayer functionalized electrodes are also a novel way to improve the contact barrier, which can significantly reduce the I_on_/I_off_ required by the device. In addition, a 2D/2D van der Waals contact is also the ideal way to optimize the contact behavior. Meanwhile, the intrinsic MoS_2_ is a typical N-type semiconductor, so it is difficult to prepare high-performance P-type MoS_2_ FETs. At the same time, the defect sites on the surface of the film will produce high resistance and hinder its electrical performance. Therefore, doping is an effective method to improve the performance of conductive channels. Metal doping and non-metal doping can improve the device mobility. More importantly, several different doping methods, such as tantalum, niobium, and He^+^ irradiation methods, can realize the transformation of a MoS_2_ FET from N-type to P-type. In addition, surface charge accumulation and leakage current due to trap charges are also key challenges for MoS_2_ FETs. A reasonable dielectric layer is significant to reduce leakage current and optimize stability. Although the high-K dielectric is a common solution, it can also produce trap charge. Doping dielectric layers, plasma treatment, double-K dielectric layers, etc., are proposed to improve the performance of the dielectric layer.

Functional 2D MoS_2_ FETs have been widely used in key and emerging fields such as logic, RF circuits, optoelectronic devices, biosensors, piezoelectric devices, and synaptic transistors. However, there are some important challenges. For example, how to obtain high-performance P-type MoS_2_ FETs on a large scale and build complementary circuits in logic circuits. The strong photoluminescence properties of the MoS_2_ material promote its application in optoelectronic devices, but its effective mechanism in photoluminescence devices and electroluminescence devices remains to be explored, and wearable high-performance biosensors are of great help to solve the current global epidemic detection. Thus, a MoS_2_ artificial synapse can be desired for real-time monitoring. Although its research has been very hot in recent years, there is still a long way to go in the application of a brain-like microarray and artificial neuromorphology. However, we still believe that molybdenum sulfide FETs can be used to construct novel functionalized devices and even super-large scale flexible electronic systems.

Two-dimensional MoS_2_ is a very suitable channel layer material for the high-performance FETs, and 2D MoS_2_ FETs are the promising candidate for downscaling electronics with short channels, low thickness, small volume, high speed, high sensitivity, light weight, etc. Especially, a MoS_2_ FET array based on a high-quality MoS_2_ channel will have a large degree of application in the next generation of integrated circuits and flexible electronics. However, monolayer MoS_2_ can better meet the high-performance requirements of short channels and even ultra-short-channel FETs. Thus, it is of great practical significance to develop the preparation process of monolayer MoS_2_ and MoS_2_ FETs performance. Thus, we believe it to be necessary to continue to extensively study in-depth fewer-layer MoS_2_ and MoS_2_-based FETs.

## Figures and Tables

**Figure 1 nanomaterials-12-03233-f001:**
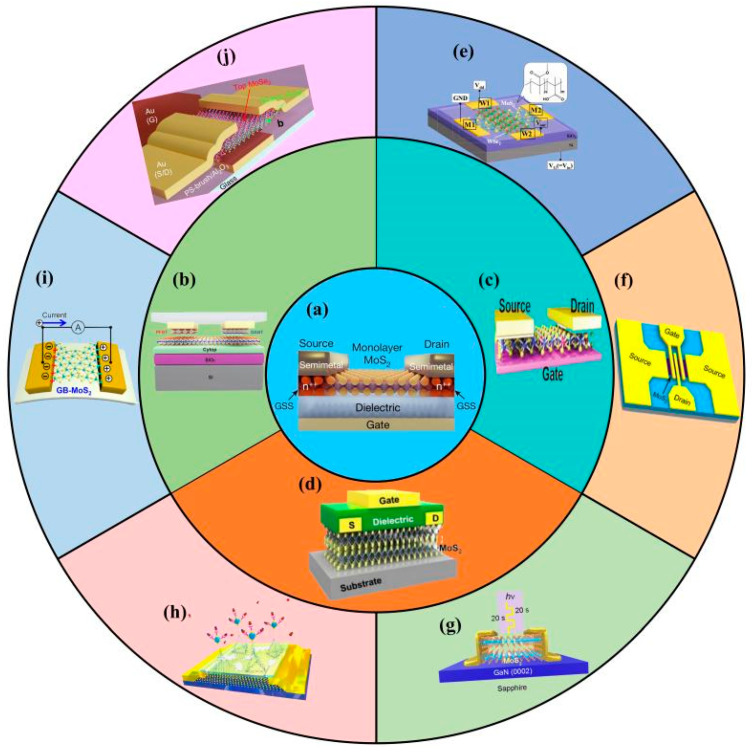
MoS_2_ FET performance improvement strategies and applications. (**a**) Schematic of monolayer MoS_2_ FET (reprinted/adapted with permission from Ref. [21]. Copyright 2021 Springer Nature). (**b**) Improving MoS_2_ FET performance by optimizing contact behavior (reprinted/adapted with permission from Ref. [14]. Copyright 2022 Wiley). (**c**) Improving MoS_2_ FET performance by regulating conductive channel (reprinted/adapted with permission from Ref. [16]. Copyright 2022 Wiley). (**d**) Improving MoS_2_ FET performance by rationalizing dielectric layer (reprinted/adapted with permission from Ref. [17]. Copyright 2020 IEEE). The applications of MoS_2_ FET in (**e**) logic circuits (reprinted/adapted with permission from Ref. [22]. Copyright 2020 American Chemical Society); (**f**) radio-frequency circuits (reprinted/adapted with permission from Ref. [23]. Copyright 2014 American Chemical Society); (**g**) optoelectronic devices (reprinted/adapted with permission from Ref. [24]. Copyright 2021 American Chemical Society); (**h**) biosensors (reprinted/adapted with permission from Ref. [25]. Copyright 2021 Elsevier); (**i**) piezoelectric devices (reprinted/adapted with permission from Ref. [26]. Copyright 2020 American Chemical Society); (**j**) synaptic transistors (reprinted/adapted with permission from Ref. [27]. Copyright 2022 American Chemical Society).

**Figure 2 nanomaterials-12-03233-f002:**
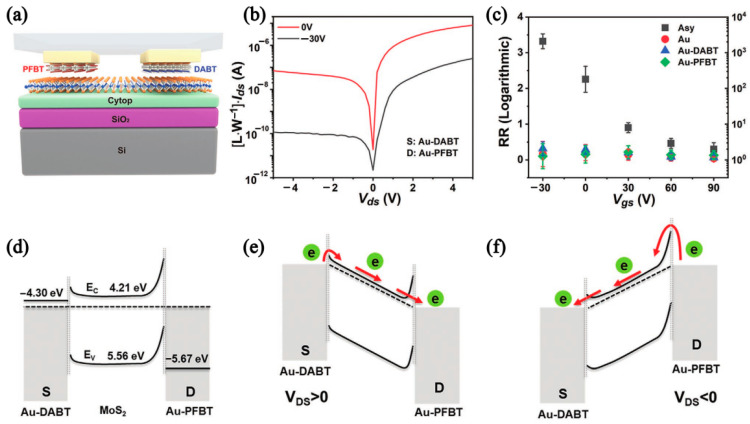
(**a**) Structure diagram of asymmetric MoS_2_ FETs. (**b**) Semilogarithmic plot of the I_ds_-V_ds_. (**c**) Rectification ratio. (**d**–**f**) Energy band diagram at thermodynamic equilibrium, positively biased source–drain, and negatively biased source–drain, respectively. (Reprinted/adapted with permission from Ref. [14]. Copyright 2022 Wiley).

**Figure 3 nanomaterials-12-03233-f003:**
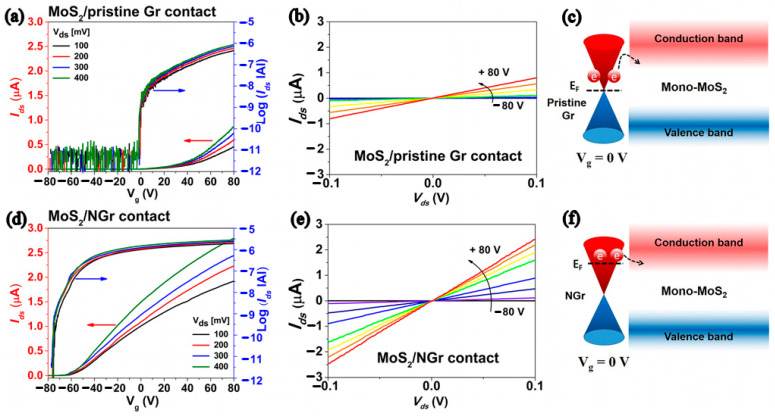
Electrical characteristics of MoS_2_/pristine Gr contact device: (**a**) Transfer characteristic (left axis) and semilogarithm (right axis). (**b**) Output characteristic. (**c**) Schematic diagram of band alignment. Electrical characteristics of MoS_2_/NGr contact device: (**d**) Transfer characteristic (left axis) and semilogarithm (right axis). (**e**) Output characteristic. (**f**) Schematic diagram of band alignment. (Reprinted/adapted with permission from Ref. [15]. Copyright 2019 American Institute of Physics).

**Figure 4 nanomaterials-12-03233-f004:**
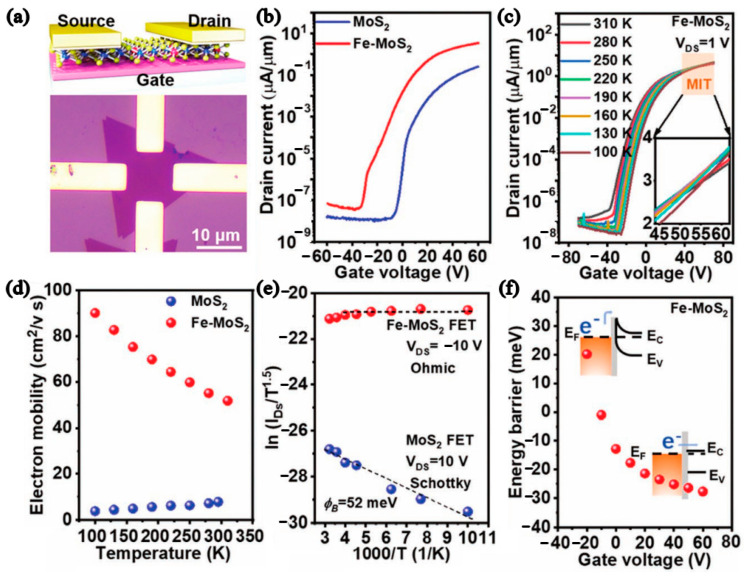
(**a**) Schematic diagram and OM graph of monolayer Fe-MoS_2_ device. (**b**) Transfer characteristic of MoS_2_-based devices. (**c**) Temperature vs. transfer characteristic. The inset shows the MIT area. (**d**) Electron mobility vs. temperature. (**e**) Arrhenius plots. (**f**) Energy barriers and energy band of the monolayer Fe-MoS_2_. (Reprinted/adapted with permission from Ref. [16]. Copyright 2022 Wiley).

**Figure 5 nanomaterials-12-03233-f005:**
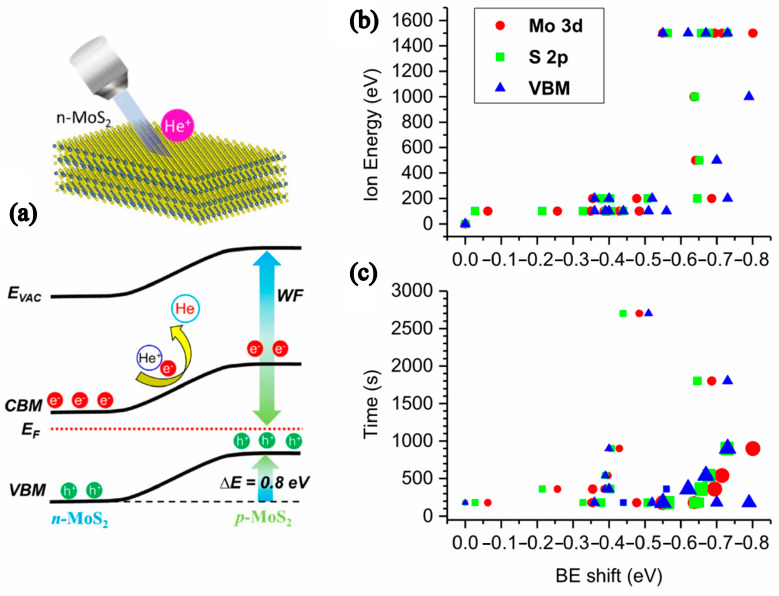
(**a**) Schematic diagram of the conversion of n-type MoS_2_. The negative binding energy shifts of Mo 3d, S 2p, and valence band photoemission spectra depend on (**b**) the ion energy band, and (**c**) irradiation time of He+ ion irradiation. (Reprinted/adapted with permission from Ref. [41]. Copyright 2022 Springer Nature).

**Figure 6 nanomaterials-12-03233-f006:**
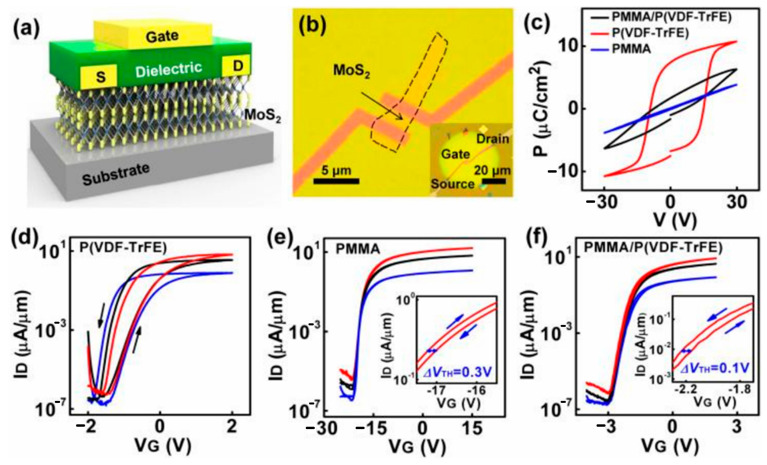
(**a**) Structure diagram of MoS_2_ transistor. (**b**) The OM graph of MoS_2_ transistor. (**c**) Ferroelectric characteristics of the different dielectrics. (**d**–**f**) Transfer properties of the MoS_2_ transistors with different gate dielectrics, at V_D_ = 1 V (Red), 0.5 V (black), and 0.1 V (blue), receptively. The insets are the enlarged figure of the transfer curve with V_D_ = 1 V. (Reprinted/adapted with permission from Ref. [17]. Copyright 2020 IEEE).

**Figure 7 nanomaterials-12-03233-f007:**
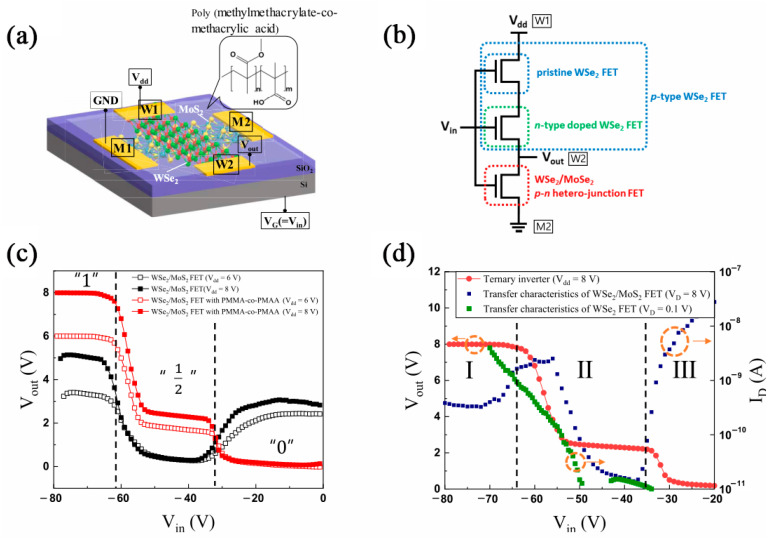
(**a**) Schematic diagram of ternary inverter by cross-type p–n heterojunction WSe_2_/MoS_2_-based FETs. (**b**) Logic circuit for ternary inverter. (**c**) V_out_ vs. V_in_ characteristic curves. (**d**) Characteristics of ternary inverter (red markers). Transfer characteristic curves of device (blue markers) and WSe_2_-based FETs (green markers). (Reprinted/adapted with permission from Ref. [22]. Copyright 2020 American Chemical Society).

**Figure 8 nanomaterials-12-03233-f008:**
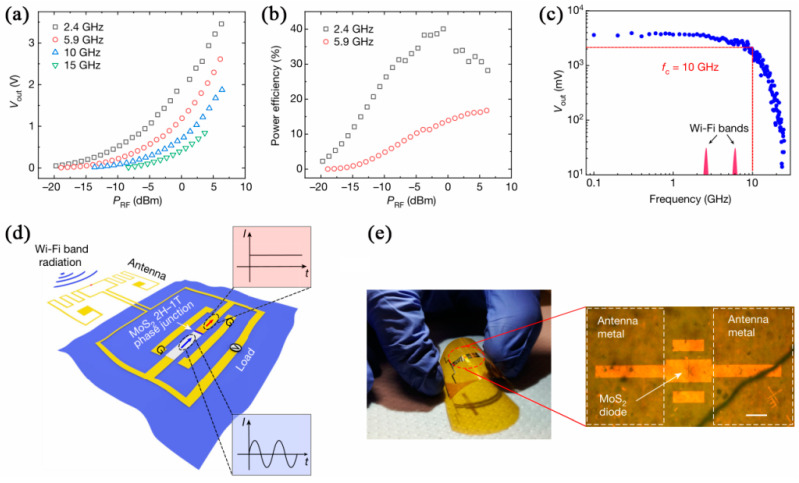
MoS_2_ phase-junction rectenna as a wireless RF energy harvester: (**a**) Output voltage vs. input RF power. (**b**) Power efficiency of MoS_2_ rectifiers vs. input power. (**c**) Output voltage vs. frequency. (**d**) Flexible MoS_2_ wireless energy harvesting. The illustrative I–t curves correspond to the a.c. (blue arrow) and d.c. (red arrow) currents. (**e**) MoS_2_ rectenna on Kapton. (Reprinted/adapted with permission from Ref. [65]. Copyright 2019 Springer Nature).

**Figure 9 nanomaterials-12-03233-f009:**
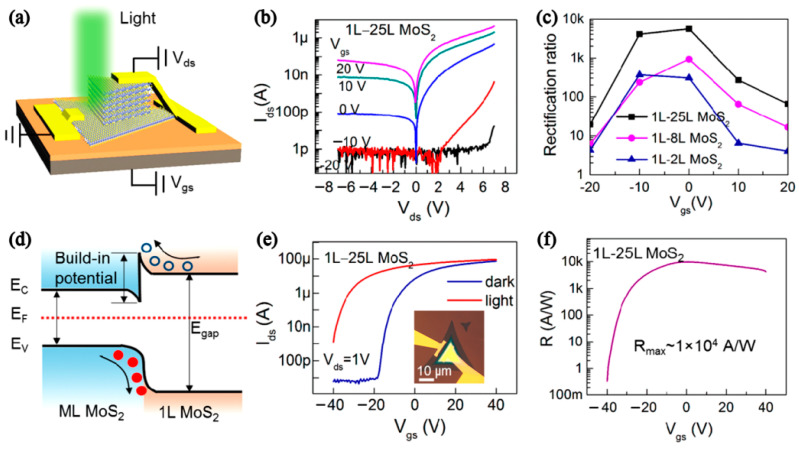
(**a**) Schematic of the monolayer−multilayer (1L–25L) MoS_2_ heterojunction device. (**b**) I_ds_-V_ds_ characteristics. (**c**) Rectification ratio as V_gs_. (**d**) Band diagram of the 1 L-ML MoS_2_ heterojunction in the off state (V_gs_ = 0; V_ds_ = 0). (**e**) I_ds_-V_gs_ characteristics. The inset is the OM graph of the device. (**f**) V_gs_ vs. photoresponsivity. (Reprinted/adapted with permission from Ref. [69]. Copyright 2019 American Chemical Society).

**Figure 10 nanomaterials-12-03233-f010:**
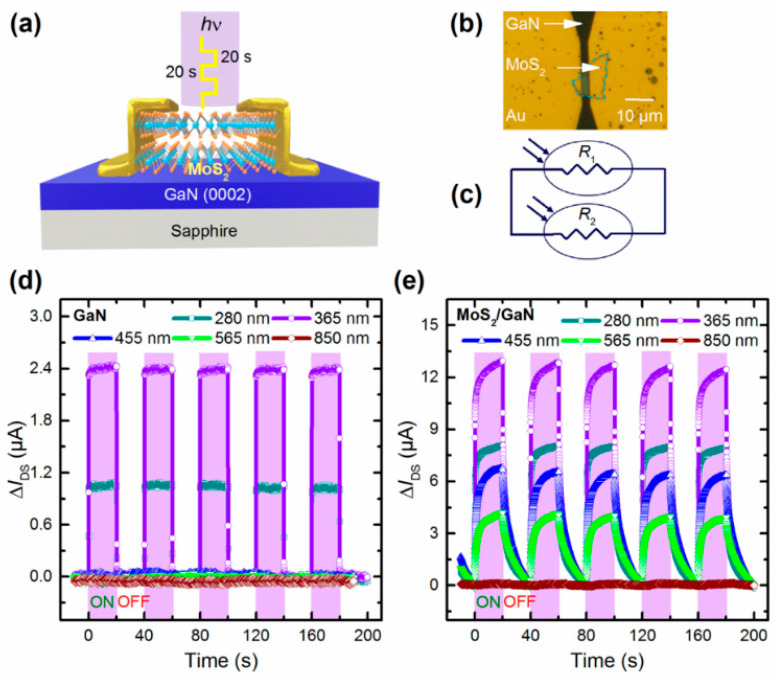
(**a**) Device structure of MoS_2_/GaN photodetector. (**b**) OM graph of the hybrid device. (**c**) Equivalent circuit diagram of the hybrid device. Photoresponse measurements of (**d**) GaN devices and (**e**) MoS_2_/GaN devices. (Reprinted/adapted with permission from Ref. [24]. Copyright 2021 American Chemical Society).

**Figure 11 nanomaterials-12-03233-f011:**
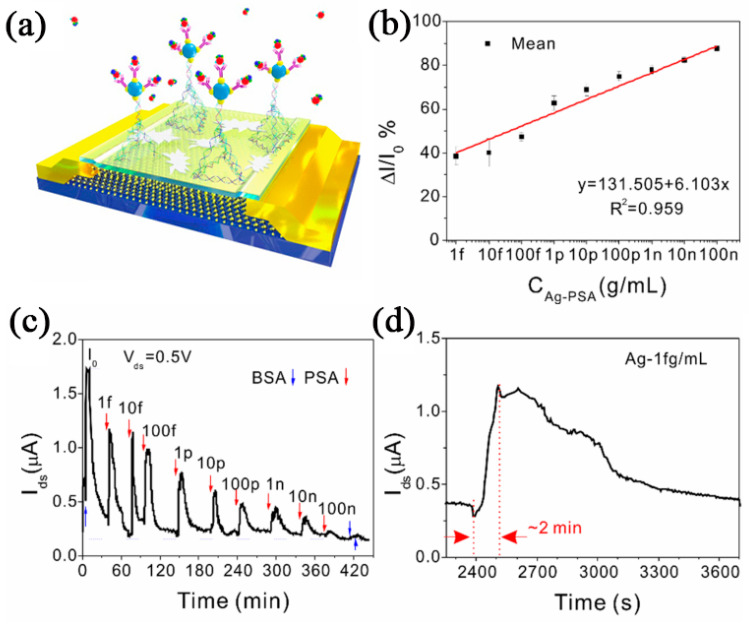
(**a**) Schematic diagram of 3D MoS_2_ biosensor. (**b**) Real-time detection (V_ds_= 0.5 V) of different concentrations. (**c**) Response time curve. (**d**) Response variation vs. PSA concentrations. (Reprinted/adapted with permission from Ref. [25]. Copyright 2021 Elsevier).

**Figure 12 nanomaterials-12-03233-f012:**
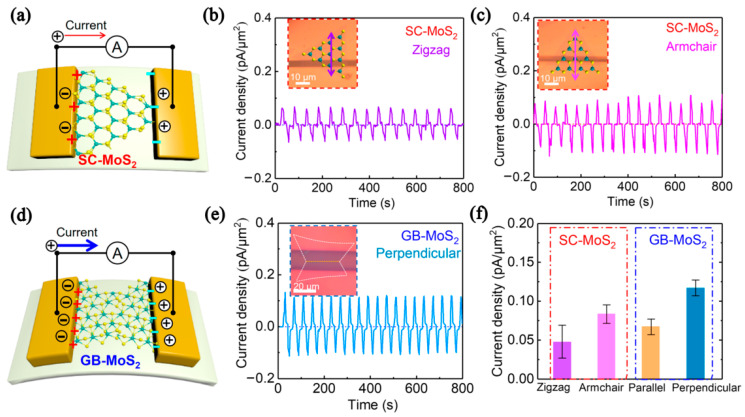
Schematic of a piezoelectric device based on (**a**) single-crystal MoS_2_ flake and (**d**) grain boundaries of MoS_2_ flake. Recorded current outputs of the SC-MoS_2_ devices are (**b**) zigzag and (**c**) armchair, and that of the (**e**) GB-MoS_2_ piezoelectric device is perpendicular. (**f**) Statistical data for the current density of different monolayer MoS_2_-based piezoelectric devices (reprinted/adapted with permission from Ref. [26]. Copyright 2020 American Chemical Society).

**Figure 13 nanomaterials-12-03233-f013:**
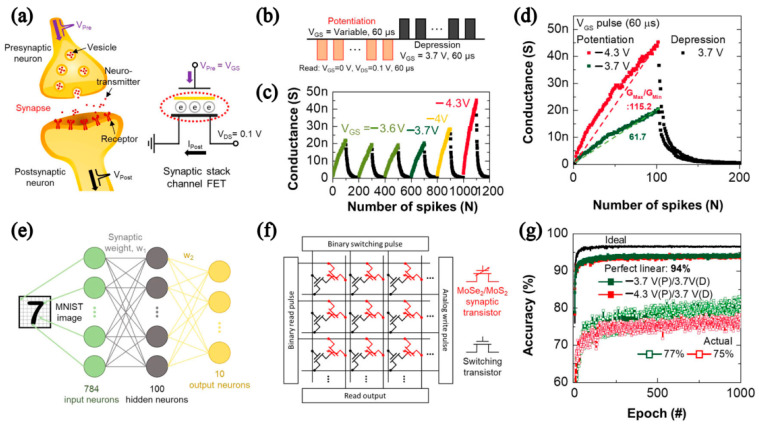
Synaptic memory behavior of stack channel FETs. (**a**) Schematic of a biological synapse in neuron system and synaptic stack channel FETs for neuromorphic function. (**b**) Illustration of a V_Pre_ pulse train. (**c**) Monitored P-D plots vs. the diverse amplitude. (**d**) Two P-D conductance plots with G_Max_/G_Min_ values. (**e**) Schematics of multilayer perceptron neural network for classification of MNIST handwritten digits. (**f**) Circuit diagram of artificial neural network. (**g**) Simulation results vs. actual P-D behavior. (Reprinted/adapted with permission from Ref. [27]. Copyright 2022 American Chemical Society).

**Table 1 nanomaterials-12-03233-t001:** Overview summary comparison of emerging 2D materials-based FETs.

Materials	Technology	Configuration	Contact Electrode	Dielectric	Mobility (cm^2^/Vs)	I_on_/I_off_	SS (mV/dec)	Ref.
MoS_2_	Exfoliation	Dual-gated	Ni/Au	Al_2_O_3_	517	10^8^	140	[7]
	CVD	Back-gated	Ti/Au	HfO_2_	118	10^8^	/	[9]
	Au-assisted Exfoliation	Back-gated	Ti/Au	SiO_2_	25	10^7^	100	[10]
	MOCVD	Top-gated	Au/Ti	SiO_2_	22	10^5^	/	[11]
	VLS	Back-gated	Au	SiO_2_	33	10^8^	980	[12]
	PLD	Top-gated	Au/Ti	HfO_2_	9	10^5^	/	[13]
	Exfoliation	Top-gated	Au	SiO_2_/Cytop	31	10^7^	/	[14]
	Exfoliation	Back-gated	Graphene	SiO_2_	9	10^6^	/	[15]
	APCVD	Back-gated	Cr/Au	SiO_2_	54	10^8^	/	[16]
	Exfoliation	Top-gated	Cr/Au	PMMA/P(VDF-TrFE)	/	10^7^	/	[17]
WSe_2_	CVD	Back-gated	Ti/Pd	BN	92	10	/	[5]
MoSe_2_	Exfoliation	Back-gated	Ni	SiO_2_	50	10^6^	/	[8]
WS_2_	Exfoliation	Back-gated	Ti/Au	SiO_2_	20	10^6^	70	[6]
BP	Exfoliation	Top-gated	Ni/Au	SiO_2_	862	10^2^	563	[4]
Graphene	CVD	Back-gated	Cr/Au	TiO_2_	1872	—	/	[3]

## Data Availability

Not applicable.

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
