# Peer review of "Evolution Application of Two-Dimensional MoS2-Based Field-Effect Transistors"

_nanomaterials, 2022, doi:10.3390/nano12183233_

Round 1

Reviewer 1 Report

Transition metal dichalcogenides (TMD) is the next important 2D material after the discovery of Graphene due to its unique electrical and optical properties. Because of its properties, the research is continuously increasing with the use of these materials in the field of electronics, optoelectronics and biosensors. Out of all, MoS2 has been the most promising material due to its band gap, high ON/OFF ratio, high carrier mobility and stability. In this paper, the authors have reviewed MoS2 based field effect transistors in key and emerging fields, including logic, RF circuits, optoelectronic devices, biosensors, piezoelectric devices, and synaptic transistors. The authors have described in detail the state of the art with key merits and limitations and prospects in the future. The authors have presented a thorough introduction to the subject with proper citations. The paper is well written and clear. However, I have few minor comments:

1) Please add a small paragraph based on common methods of growth of MoS2 (via CVD) and extraction of MoS2 single layer (Exfoliation and transfer) with the scale i.e, how big MoS2 can be grown by each method.  

2) The authors have used inconsistent language in the work scitation. Instead of presenting the university, where the work has been done, can you please change to for eg: ‘Samori et al or Samori and co workers’ instead of Samori from University of Strasbourg in line 103. This needs to be corrected at several places throughout the paper. 

3) Surface accumulation of charges and charge traps on dielectric are big issues leading to high leakage current. It has been experimentally observed using STM as shown by: https://doi.org/10.1038/s41467-018-03824-6, please cite this information.

Thank you 

Priyanka

Reviewer 2 Report

The review by Wang et al. titled “Green Production of Planar Aligned Dense 2D Nano-oxides on CNT paper by Ultrafast Laser Induced High-Pressure Photochemistry for Stable High-rate LIB Anode Evolution Application of two-dimensional MoS2 -Based Field Effect Transistorss” attempts to discuss MoS2 TMD materials for applications in photodetectors, biosensors, piezoelectric devices, and transistors.  Within the article, contact barriers, channel properties have also been discussed. Overall, the work is suitable for this journal. However, a few things need to be done prior for consideration.

1.    There are various other 2D materials. The authors should tell the reader why they selected and focused this review on MoS2. This has not been provided.

2.    Under section 2. “MoS2 field effect transistors” The author should clearly explain how FETs work, and why MoS2./ How can they compare with Si  or other 2D materials based FETs, etc? A table of performance can be thought of. Also, different MoS2 FETs perform different, as per literature studies, why? And how to improve

3.    In section 2.1. Contact barrier. It is ultimately important for the authors to read and consider this reference (Shen et al., Nature 2021, 593 (7858), 211-217). From this, several ultra low resistance contacts should be discussed both 2D and non 2D.

Round 2

Reviewer 2 Report

The authors carefully addressed all the comments from the reviewers. I would recommend accepting the article.